# Effects of Aging and Methionine Restriction on Rat Kidney Metabolome

**DOI:** 10.3390/metabo9110280

**Published:** 2019-11-14

**Authors:** Irene Pradas, Mariona Jové, Rosanna Cabré, Victoria Ayala, Natalia Mota-Martorell, Reinald Pamplona

**Affiliations:** Department of Experimental Medicine, Lleida University-Institute for Research in Biomedicine of Lleida (UdL-IRBLleida), 25198 Lleida, Spain; ipradas@mex.udl.cat (I.P.); mariona.jove@udl.cat (M.J.); Rosanna.cabre@mex.udl.cat (R.C.); victoria.ayala@mex.udl.cat (V.A.); natalia.mota@mex.udl.cat (N.M.-M.)

**Keywords:** aging, dietary restriction, mass spectrometry, lipidomics, protein damage, renal cortex

## Abstract

Methionine restriction (MetR) in animal models extends maximum longevity and seems to promote renoprotection by attenuating kidney injury. MetR has also been proven to affect several metabolic pathways including lipid metabolism. However, there is a lack of studies about the effect of MetR at old age on the kidney metabolome. In view of this, a mass spectrometry-based high-throughput metabolomic and lipidomic profiling was undertaken of renal cortex samples of three groups of male rats—An 8-month-old Adult group, a 26-month-old Aged group, and a MetR group that also comprised of 26-month-old rats but were subjected to an 80% MetR diet for 7 weeks. Additionally, markers of mitochondrial stress and protein oxidative damage were analyzed by mass spectrometry. Our results showed minor changes during aging in the renal cortex metabolome, with less than 59 differential metabolites between the *Adult* and *Aged* groups, which represents about 4% of changes in the kidney metabolome. Among the compounds identified are glycerolipids and lipid species derived from arachidonic acid metabolism. MetR at old age preferentially induces lipid changes affecting glycerophospholipids, docosanoids, and eicosanoids. No significant differences were observed between the experimental groups in the markers of mitochondrial stress and tissue protein damage. More than rejuvenation, MetR seems to induce a metabolic reprogramming.

## 1. Introduction

The physiological aging process involves a loss-of-function of many organs, and the kidney, which has one of the highest rates of susceptibility to age-dependent tissue damage, is no exception. In fact, the most common cause of death in male rats is renal failure [1]. In terms of morphological alterations, the average weight of kidneys progressively decreases with aging, affecting mainly the renal cortex. The total nephron size and number also decline with aging, and there is a global increase in glomerulosclerosis, tubular atrophy, and interstitial fibrosis [2]. It has been suggested that these changes are associated with an age-related increase in lipid accumulation of triacylglycerides (TAG) and cholesterol content, likely due to changes in the renal expression of transcriptional factor sterol regulatory element binding protein-1 and -2 (SREBP-1 and SREBP-2) [3]. Interestingly, lipid accumulation potentially leads to lipid peroxidation and, with advancing age, there is an increase in renal free radical production along with a decrease in the expression of antioxidant enzymes [4,5,6]. In this line, levels of F2-isoprostanes are found to be increased in the kidney of old rats [7].

In terms of kidney function, the glomerular filtration rate diminishes with advancing age, as does renal tubular function. Consequently, the ability of the kidney to both maximally concentrate and dilute urine diminishes and the handling of sodium and potassium is also affected [1,8]. Lipid metabolism has been found to be closely related to the age-related decline in renal function through the role of arachidonic acid (AA) and its metabolites [9]. Importantly, this finding is supported by the accumulation of lipid species rich in AA that are found in the lipidomic profile of the kidney [10]. An increase in long-chain ceramides and lysophosphatidylcholine (LPC) species, and a decrease in phosphatidylcholine (PC) species, all potentially related to the development of renal fibrosis, has been reported in the aged rodent kidney [11]. A decrease in AA and its metabolites, epoxyeicosatrienoic acid (EET) and dihydroxyeicosatrienoic acid (HETE), generates an imbalance between vasodilatory and vasoconstrictive events affecting renal function in aging kidneys [12,13,14].

Successful aging relies upon a nutritionally complete diet and adequate exercise to counteract some of the physiologic changes related to age. In fact, dietary restrictions extend both median and maximum lifespans in several animal species, reduce oxidative stress and resulting molecular damage, and remodel lipid metabolism into a healthier lipid profile [15]. In this context, previous studies have demonstrated that dietary methionine restriction (MetR), without changes in energy intake, can increase maximum lifespan in different experimental models [15,16,17]. It has also been suggested that MetR may have a renoprotective effect in aged rodents—Possibly through the action of hydrogen sulfide in the trans-sulfuration pathway [18]. The application of MetR induces lower mitochondrial free radical production, and decreases mitochondrial DNA damage and the levels of specific markers of mitochondrial protein damage of the rat heart and liver, as well as brain and kidney [19,20,21,22,23]. It has also been reported that MetR is able to reduce the incidence of age-associated degenerative diseases and improve glucose metabolism by decreasing insulin, insulin-like growth factor 1 (IGF-1), and glucose levels in blood [24]. Furthermore, MetR induces weight loss along with a reduction of fat deposition, especially in visceral and ectopic depots [25,26,27]. Finally, MetR reduces plasma and hepatic TAGs and cholesterol levels, while lowering and increasing leptin and adiponectin plasma levels respectively [27,28,29,30,31,32]. Supporting these biochemical adaptations to MetR, several studies in adult and young rodents have shown that this anti-aging nutritional intervention induces significant changes in lipid-related gene and protein expression in a tissue-specific manner, leading to different fatty acid profiles in adipose tissue, liver, skeletal, and cardiac muscle and the brain [23,28,29,31,32,33,34,35].

All the studies published to date, which have reported MetR modification of different metabolic pathways, were performed using young immature rodents. It is currently unknown if MetR can also modify the metabolic profile in aged animals, or whether it can reverse the putative effects of aging. In particular, it is important to know if a short-term intervention of MetR at old age is beneficial in animals that have been feeding ad libitum during all their previous lifespan. With that aim, three groups were established in the study design—Adult (8 months old, control diet), Aged (26 months old, control diet), and MetR (26 months old, 80% MetR diet). Since MetR has a profound effect on metabolism, liquid chromatography-mass spectrometry (LC-MS)-based untargeted metabolomic and lipidomic analyses of renal cortex samples of the three groups were performed. Markers of mitochondrial stress and tissue protein oxidative damage were also detected and measured by gas chromatography-mass spectrometry (GC-MS).

## 2. Results

### 2.1. Effects of Aging and MetR on Markers of Mitochondrial Stress and Tissue Protein Damage

No significant differences were observed between the Adult and Aged groups, nor between the Aged and MetR groups, in the marker of mitochondrial stress 2-S-(2-succino)cysteine [2-SC], and different specific markers of protein oxidation (the protein carbonyls glutamic [GSA] and aminoadipic [AASA] semialdehydes), glycoxidation (carboxyethyl-lysine [CEL]), lipoxidation (malondialdehyde-lysine [MDAL]), and mixed glyco- and lipoxidation (carboxymethyl-cysteine [CMC] and carboxymethyl-lysine [CML]) (Table 1).

### 2.2. Effect of Aging on the Renal Cortex Metabolome

An untargeted lipidomic and metabolomic approach to define and compare the renal cortex metabolome of the Adult and Aged groups was designed, and the data obtained were subjected to an unsupervised multivariate statistical analysis. After filtering those molecules present in at least 50% of the samples of the same group, we obtained 528 species from lipidomics and 839 species from metabolomics approaches. The principal component analysis (PCA) performed using these molecules revealed no clear differences between the Adult and Aged experimental groups. Illustrative images of the multivariate statistics are shown in Figure 1. The first component of the PCA showed only slight differences between the two groups, with both principal components explaining 17.6% of the variability across the samples for the lipidomic analysis. In contrast, when hierarchical clustering analysis was applied, we obtained a better classification of the samples according to the animal age.

Moreover, when a comparison was made of the 25 lipid species with the highest statistically significant differences between the Adult and Aged groups, using a hierarchical clustering analysis (represented in the heat map of Figure 1), differences were observed in the lipid profiles of the two groups. The non-parametric t-test for equal variances that was performed found that 28 compounds detected from lipidomics and 31 from metabolomics methodology were statistically significantly different between the groups. Twelve out of these were identified (Table 2). All the un-identified compounds obtained by untargeted lipidomics and metabolomics are represented in Appendix A. Among the identified molecules we found almost all the glycerolipid (GL) species and one phosphatidylcholine (PC) ether lipid up-regulated with advancing age. Furthermore, ADP-ribose and NADH were significantly decreased in aged animals. Interestingly, a few oxylipins (some of which derive from AA) that down-regulated in the Aged group were also identified.

### 2.3. Effect of MetR on the Renal Cortex Metabolome

The evaluation of the effect of the MetR diet at old age on the renal cortex of rats was also based on the untargeted metabolomic and lipidomic analyses that were performed. The unsupervised PCA and the hierarchical clustering analysis showed no clear differences between the three experimental groups (Adult, Aged, and MetR) for either the lipidomic and metabolomics approaches. Illustrative images of the multivariate statistics are shown in Figure 2. In line with this, hierarchical clustering analysis also showed no differences between the groups. However, when a comparison was made of the 25 compounds with the highest statistically significant differences between the three experimental groups, using a hierarchical clustering analysis (represented in the heat map of Figure 2), clear differences were observed.

In order to identify those entities responsible for a specific lipid/metabolic profile in the kidney, a one-way ANOVA was performed with a post hoc Tukey’s test to compare the three experimental groups. Out of the 583 lipid species that were detected and filtered in the untargeted lipidomic analysis, 46 were found to be statistically significantly different between the groups, and 15 of these were identified (Table 3). Among the 925 metabolites that were detected and filtered in the untargeted metabolomic analysis, 71 were found to be statistically significantly different between the groups, and 15 of these were also identified (Table 4). All the non-identified compounds obtained by untargeted lipidomic and metabolomic analysis are shown in Appendix A. Most of the identified lipid species are glycerophospholipids (GPs), although there are also some sphingolipids (SPs) and GLs. Of the SP compounds, only two species were identified, a sphingomyelin (SM) and a derivative of a hexosylceramide (HexCer). In the MetR group, levels of SM and HexCer were upregulated and downregulated, respectively, with respect to both the Aged and the Adult groups. In addition, different types of GLs were identified—monoacylglycerides (MAG), diacylglycerides (DAG), and TAG species. All of them presented polyunsaturated fatty acid (PUFA) chains and were upregulated in the MetR group with respect to both the Aged and the Adult group. However, regulation in the MetR group with respect to the Aged control group was different for each lipid, and no pattern was detected. Finally, 50% of the PC species that were identified were upregulated in the MetR group with respect to both the Aged and the Adult group, and the other 50% downregulated. Finally, the identified lysoglycerophospholipids (LGPs) appeared to be regulated in a class-dependent way (being the LPE increased and LPI decreased in MetR group, respectively), whereas the identified ether lipids, phosphatidic acid PA (P-30:1) and phosphatidylserine PS(P-29:0), were upregulated due to the effect of the MetR diet.

Different regulation effects were found in the species that were detected and identified with the metabolomic approach. 10,11-DiHDPE and 20-carboxy-LTB4 were up- and down-regulated, respectively in the MetR group with respect to both the Adult and Aged groups. With respect to purine metabolism, inosine was up-regulated and ADP-ribose was down-regulated in the MetR group with respect to the Aged group. 5-methylcytidine was down-regulated in the MetR group with respect to the Aged group. The renal cortex fatty acyls that were conjugated with coenzyme A or acylcarnitine species were also altered in the MetR group, mostly upregulated with respect to both the Adult and Aged control groups. Bile alcohol and a bile acid were also identified, the latter compound was upregulated in the MetR group with respect to both the Adult and Aged control groups.

Finally, glutathione disulfide (GSSG) and a prostaglandin were found to be downregulated in the MetR group with respect to both the Adult and Aged control groups.

## 3. Discussion

The physiological aging process requires adaptation for the maintenance of metabolic homeostasis at every biological level (molecular, cellular, organic, and systemic), having to support any imbalance or shift in the metabolism that can self-intensify, and ultimately clinically manifest itself [36]. The up- and down-regulation of metabolic pathways provide a specific metabolome for each type of tissue, and so age-related changes can vary from one tissue to another [10].

In recent years, different studies have performed untargeted metabolomic analyses of plasma and serum to elucidate the metabolic changes produced with advancing age in both humans and rodents, demonstrating alterations in lipid homeostasis and one carbon metabolism [37,38,39,40,41,42,43,44]. With respect to lipid homeostasis, studies performed on 23- and 26-month-old C57BL/6 mice revealed the existence of a metabolomic signature associated with aging that involves changes in acylcarnitines and a decrease in plasma levels of LPC, SM, and cholesterol derivatives [37,41], while cholesterol plasma levels were significantly increased by the aging process in 26-month-old male Wistar rats [45]. Further metabolomic studies on 117 wild-type mice of different genetic backgrounds ranging from 8–129 weeks of age, reported a decrease in LPC and SM species and defined phosphatidylethanolamine PE (16:0/18.2) and PE (16:0/22:6) as biomarkers that increased with the aging process, while ether PE (P-) plasma levels decreased [40]. Metabolomic studies on human plasma confirmed that SP and GP species, including LGP and ether lipid species, could be critical candidate metabolites in the aging process [38,39,42,43,44]. In addition, these studies point to the importance of acylcarnitine and the β-oxidation process due to an age-related plasma level increment [38,42].

The untargeted metabolomic and lipidomic analyses of the renal cortex in the present study demonstrate specific changes associated with the aging process in the metabolome and lipidome of the analyzed tissue, with lipid metabolism being especially significant. Differences in structural lipids (GPs), bioactive lipids (oxylipins), and lipid species involved in energy metabolism (acylglycerols) were found. Initially, the unsupervised multivariate statistical analysis, applied to the results of the untargeted metabolomic and lipidomic analyses, revealed no differences between the groups, suggesting that the renal cortex variability with respect to the whole metabolome cannot be explained by the aging process. However, when a set of differential metabolites was selected, aging was found to be a key factor that separated the groups, suggesting that this physiological process affects particular metabolic pathways or even specific reaction steps.

Specifically, in the analyzed renal metabolome, approximately 4% of the polar and non-polar metabolites were statistically significantly different between the experimental groups. The aging process in renal tissue did not affect any GP species, except for an increase in PC (P-) species. This age-associated regulation of ether lipids is in agreement with previous metabolomic studies performed on mice and human plasma [39,40]. The physiological role of plasmalogens is essentially linked to their function as membrane components, although other specific functions (as second messenger and antioxidant components) have been ascribed to them [46,47]. Further studies need to be performed to elucidate the potential meaning of these changes.

Other structural lipid species such as SP and cholesterol remained unaltered in the renal cortex during the aging process. The role of SP and cholesterol in biological membranes is the formation of lateral lipid clusters, denominated lipid rafts, in which particular molecules are concentrated and participate in membrane-mediated signaling events. The high affinity of SM and cholesterol locates them together in these microdomains, where the SM concentration seems to control the cholesterol distribution, and on the surface of the lipoproteins [48]. Furthermore, HexCer helps to stabilize the membranes, and their complex products, gangliosides, modulate the charge density at the membrane surface [49,50]. An absence of changes in the composition of structural lipids in the renal cortex means that their physical properties and function in membranes are maintained during the aging process.

Some AA-derived oxylipins were decreased in the Aged group compared to the Adult group. Little has been reported in the literature about the physiological roles of oxylipins and the effect of aging on their metabolism in mammalian tissues. A recent study provided the first comprehensive profile of oxylipins in the kidney, liver, and serum of adult healthy rats, demonstrating that oxylipin profiles can be highly divergent across mammalian tissues [51]. The results of the present study showed a decrease with age in renal AA and its metabolites, which is in agreement with previous published studies [12,13,14]. Thus, the age-related decline in renal function could be due to a decrease in AA and its metabolites that can generate an imbalance between vasodilatory and vasoconstrictive events affecting the renal function in aging kidneys.

As for lipid involvement in energy metabolism, aging affected the levels of some acylglycerol species. The renal cortex of the Aged group showed an increase in MAG (18:0) and DAG with PUFA chains. However, the identified TAG species were found to be both decreased and increased with aging, depending on the individual lipid species. The increase in acylglycerols with PUFA chains agrees with previous studies in rodents, where an association of higher levels of MAG, DAG, and TAG seems to be due to an increase in lipid synthesis, maybe through sterol regulatory element-binding proteins (SREBP) [3].

A MetR diet has been shown to not only improve biomarkers of metabolic health and increase longevity in rodents, but also to produce a highly integrated series of physiological and biochemical responses, including remodeling lipid metabolism into a healthier lipid profile [52]. Several studies have focused on the effects of MetR on lipid metabolism, showing differences across tissues with coordinated remodeling of lipid metabolism in liver, skeletal muscle, and adipose tissue—increasing lipogenic expression in adipose tissue while decreasing it in liver and skeletal muscle, thus favoring fatty acid oxidation [23,29,31,32,33,34,35,52,53,54,55,56]. This effect on total fatty acid composition also affects chain length and the degree of unsaturation of fatty acids from GP, TAG, and cholesterol esters (CE) both circulating or from the liver, kidney, heart, brain, and skeletal muscle under dietary restriction, suggesting a metabolic reprogramming that involves an up-regulation of β-oxidation and lower level products of oxidative damage [7,29,57,58,59,60,61,62,63].

Our study used 8- and 26-month-old rats to evaluate the effect of a 7-week diet with methionine restriction (MetR) on the metabolome and lipidome of aged animals. The number of significantly altered metabolites and lipid species in the renal cortex of rats subjected to MetR corresponds to approximately 7.75% of their metabolome. The effect of MetR led to specific remodeling of their metabolomic and lipidomic profiles. Globally, the differences between the aged animals with and without MetR strongly indicate that the metabolomic pattern is altered by this dietary intervention. The differences found in the metabolomic and lipidomic profiles of the 8-month-old adult rats compared to those of the aged rats subjected to MetR suggest that, while it may not result in a reversal of the aging-associated metabolic state, this dietary restriction does result in a remodeling/reprogramming of these profiles.

## 4. Materials and Methods

Animals and diets. Thirty male Wistar rats of 430–670 g body weight were individually caged and maintained in a 12:12 (light:dark) cycle at 22 ± 2 °C and 50 ± 10% relative humidity. Ten 8-month-old rats were fed with a control diet (Adult control group), ten 26 months-old rats were also fed with a control diet (Aged control group), and the final ten, also 26 months old, were fed with an 80% L-methionine restricted diet (MetR group). The animals were fed ad libitum with a semi-purified diet prepared by MP Biochemicals (Irvine, CA, USA). The composition of the control diet (in g/100 g of diet) was—L-arginine 1.12, L-lysine 1.44, L-histidine 0.33, L-leucine 1.11, L-isoleucine 0.82, L-valine 0.82, L-threonine 0.82, L-tryptophan 0.18, L-methionine 0.86, L-glutamic acid 2.70, L-phenylalanine 1.16, L-glycine 2.33, dextrine 5.0, corn starch 31.82, sucrose 31.79, cellulose 5.0, choline bitartrate 0.20, MP vitamin diet fortification mixture 1.0, mineral mix (AIN) 3.50, and corn oil 8.0. The composition of the methionine restriction diet was similar to that of the control diet except that L-methionine was present at 0.17% instead of the 0.86% of the control diet. The 80% reduction in L-methionine in the restriction diet was compensated with increases in the other dietary components in proportion to their presence in the diet. The two control groups received the same amount of food everyday as the MetR group had eaten on average the previous week (pair feeding). After 7 weeks of dietary treatment, the animals were sacrificed by decapitation. The renal cortex samples were immediately processed or frozen at −80 °C for subsequent assays. All procedures followed the protocols approved by the Catalonian Institutional Committee of Care and Use of Animals (Spain) and the experiments were approved by the Ethics Committee of the University of Lleida (EAEC 18-01/12).

Markers of mitochondrial stress and tissue protein damage measured by GC-MS. The marker of mitochondrial stress 2-S-(2-succino)cysteine [2-SC], and different specific markers of protein oxidation (the protein carbonyls glutamic [GSA] and aminoadipic [AASA] semialdehydes), glycoxidation (carboxyethyl-lysine [CEL]), lipoxidation (malondialdehyde-lysine [MDAL]), and mixed glyco- and lipoxidation (carboxymethyl-cysteine [CMC] and carboxymethyl-lysine [CML]) were determined by gas chromatography-mass spectrometry (GC-MS) in renal cortex samples, as previously described [64]. Markers were determined as trifluoroacetic acid methyl ester (TFAME) derivatives using an HP6890 Series II gas chromatograph (Agilent Technologies, Barcelona, Spain) with an MSD5973A Series and a 7683 Series automatic injector, an Rtx-5MS Restek column (30 mL * 0.25 mm ID * 0.25 μm particle size), and the described temperature program [64]. Quantification was performed by internal and external standardization using standard curves constructed from mixtures of deuterated and non-deuterated standards. Analyses were carried out by selected ion-monitoring GC/MS (SIM-GC/MS). The ions used were—lysine and [2H8] lysine, m/z 180 and 187, respectively; 5-hydroxy-2-aminovaleric acid and [^2^H_5_] 5-hydroxy-2-aminovaleric acid (stable derivatives of GSA), m/z 280 and 285, respectively; 6-hydroxy-2-aminocaproic acid and [^2^H_4_] 6-hydroxy-2-aminocaproic acid (stable derivatives of AASA), m/z 294 and 298, respectively; CEL and [^2^H_4_] CEL, m/z 379 and 383, respectively; MDAL and [^2^H_8_] MDAL, m/z 474 and 482, respectively; CML and [^2^H_2_] CML, m/z 392 and 394, respectively; CMC and [^13^C_3_-^15^N] CMC, m/z 271 and 273, respectively; and 2-SC and [^2^H_2_] SC, m/z 284 and 286, respectively. The amounts of product were expressed as μmoles of GSA, AASA, CEL, MDAL, CML, CMC, or SC per mole of lysine. A one-way ANOVA with post hoc Tukey’s test was performed with RStudio statistical software (RStudio, Inc., Boston, MA, USA).

Tissue homogenization. For untargeted lipidomic analysis, 20 volumes of cold homogenization buffer (180 mM of potassium chloride (KCl), 5 mM of 3-(N-morpholino)propanesulfonic acid (MOPS), 2 mM of ethylenediamine tetra-acetic acid (EDTA), 1 mM of diethylenetriamine penta-acetic acid (DTPA), and 1 µM of 2,6-Di-tert-butyl-4-methylphenol (BHT) adjusted to pH 7.4) were added to 50 mg of tissue. For the untargeted metabolomic analysis, 10 volumes of cold phosphate-buffered saline solution (pH 7.47) were added to 50 mg of tissue. All samples were homogenized at 4 °C with an Ultra-Turrax (IKA, Staufen, Germany) homogenizer.

Untargeted lipidomic analysis. An untargeted lipidomic analysis was performed using an Agilent 1290 ultra-performance liquid chromatograph (UPLC) coupled to an ESI-QTOF MS/MS 620 model (Agilent Technologies, Barcelona, Spain) as previously described [35]. For lipid extraction, 10 µL of the homogenized tissue was mixed with 5 µL of Milli-Q water and 20 µL of ice-cold methanol. The samples were vigorously shaken by vortexing for 2 min, and then 250 µL of methyl tert-butyl ether (MTBE), containing internal lipid standards, were added. The samples were immersed in a water bath (ATU Ultrasonidos, Valencia, Spain) with an ultrasound frequency and power of 40 kHz and 100 W, respectively, at 10 °C for 30 min. Then, 25 μL of Milli-Q water were added to the mixture, and the organic phase was separated by centrifugation (1400* *g*) at 10 °C for 10 min [65]. The lipid extracts, contained in the upper phase, were collected and subjected to mass spectrometry. A pool of all lipid extracts was prepared and used as quality controls. The internal lipid standards used were isotopically labeled lipids (Appendix A). Stock solutions were prepared by dissolving lipid standards in MTBE at a concentration of 1 mg/mL, and working solutions were diluted to 2.5 μg/mL in MTBE. The lipid extracts were analyzed following a previously published method [66]. The sample compartment of the UPLC was refrigerated at 4 °C and, for each sample, 10 μL of lipid extract was injected into a Waters Acquity HSS T3 column (150 mm L, 2.1 mm ID, 1.8 µm particle size) (Waters, Milford, MA, USA) heated to 55 °C. The flow rate was 400 μL/min, with solvent A consisting of 10 mM ammonium acetate in acetonitrile-water (40:60, v/v) and solvent B of 10 mM ammonium acetate in acetonitrile-isopropanol (10:90, v/v). The gradient of solvent B went from 40% at the start to 100% at the end after 10 min, followed by a two minute hold. Finally, the system was switched back to 60% B and was equilibrated for 3 min.

Untargeted metabolomic analysis. The tissue samples (40 mg) were homogenized in cold methanol (20 v/w) containing 1 μg/mL of phenylalanine C13 as the internal standard and 1 μM butylhydroxytoluene as the antioxidant, obtaining a final concentration of 50 mg tissue/mL. Then, samples were incubated at 20 °C for 1 h and centrifuged at 12,000**g* for 3 min. The supernatants were analyzed by liquid chromatography using an Agilent 1290 system coupled to an ESI-Q-TOF MS/MS 6520 model (Agilent Technologies, Barcelona, Spain). In all cases, 2 μL of the extracted sample were injected into a Zorbax SB-Aq reverse-phase column (50 mm * 2.1 mm ID, 1.8 μm particle size) (Agilent Technologies, Barcelona, Spain) equipped with a Zorbax-SB-C8 rapid resolution cartridge precolumn (2.1 mm * 30 mm ID, 3.5 μm particle size) (Agilent Technologies, Barcelona, Spain) with a column temperature of 60 °C. The flow rate was 0.6 mL min−1. Solvent A was composed of water containing 0.2% acetic acid, and solvent B of methanol 0.2% acetic acid. The gradient of solvent B went from 2% at the start to 98% at the end after 13 min, followed by a 6 min hold at 98%. Post-time was established in 5 min, as previously described [67].

For both untargeted approaches, metabolomics and lipidomics, duplicate runs of the samples were performed to collect positive and negative electrospray ionized (ESI) lipid species in a TOF mode, operated in full-scan mode at 100 to 3000 m/z in an extended dynamic range (2 GHz), using N_2_ as the nebulizer gas (5 L/min, 350 °C). The capillary voltage was set at 3500 V with a scan rate of 1 scan/s. Continuous infusion using a double spray with masses 121.050873, 922.009798 (positive ion mode) and 119.036320, 966.000725 (negative ion mode) was used for in-run calibration of the mass spectrometer.

Data Analyses and Statistics. Data obtained from the untargeted lipidomic and metabolomic analysis were acquired using MassHunter Data Analysis software (Agilent Technologies, Barcelona, Spain). The molecular feature extractor (MFE) algorithm of the MassHunter Qualitative Analysis software (Agilent Technologies, Barcelona, Spain) was used to obtain the molecular features of the samples, representing different, co-migrating ionic species of a given molecular entity (i.e., ion adducts) [68]. This algorithm used the accuracy of the mass measurements to group related ions (basing on charge-state envelope, isotopic distribution and/or the presence of different adducts and dimers/trimers) assigning multiple species (ions) to a single compound referred to as a feature. Finally, the MassHunter Mass Profiler Professional software (Agilent Technologies, Barcelona, Spain) was used to perform untargeted analysis of the extracted features. Only common features (found in at least 50% of the samples of the same condition) were taken into account to correct for individual bias. Multivariate statistical analyses (hierarchical clustering and PCA) were performed using both MassHunter Mass Profiler Professional and MetaboAnalyst software [69,70]. The masses there were found to have significant differences by ANOVA (*p* < 0.05 with Benjamini-Hochberg multiple testing correction) were searched against the Human Metabolome Database [71] and the LIPID MAPS database (Lipidomics Gateway, http://www.lipidmaps.org/, May 2014) (exact mass ppm < 20), and the MS/MS spectra were checked using the LipidBlast software [72] and MassBank Database (massbank.jp/Search). For a better identification, the retention time of the metabolites and lipid species were confirmed using authentic standards.

## Figures and Tables

**Figure 1 metabolites-09-00280-f001:**
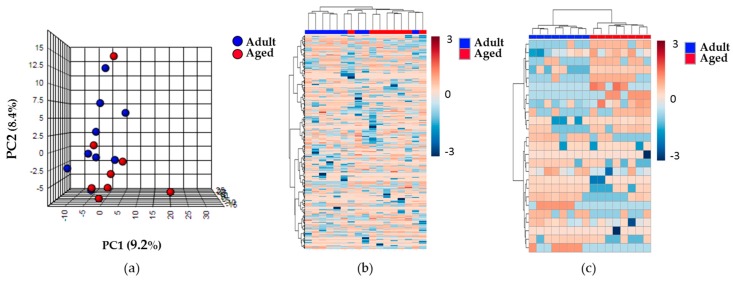
Lipidomic profile of the renal cortex samples of the Adult and Aged experimental groups. (**a**) The PCA shows no clear differences between the two experimental groups. (**b**) Heat map representation of the 528 compounds found in renal cortex. (**c**) Heat map representation of the hierarchical clustering of the 25 lipid species with lower *p*-values by a non-parametric *t*-test. Each line represents an accurate mass ordered by retention time, colored by its abundance intensity, normalized to internal standard, and baselined to median of all samples. The scale from -3 (blue) to 3 (red) represents this normalized abundance in arbitrary units. For all of them, data was acquired in the untargeted lipidomic profiling analysis based on LC-MS with ESI (+).

**Figure 2 metabolites-09-00280-f002:**
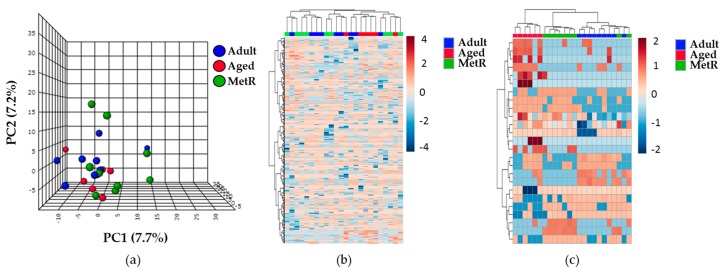
Lipidomic profiles of the renal cortex samples from animals of the three different experimental groups. (**a**) The PCA reveals no clear differences between the groups in the lipidomic profile. (**b**) Heat map representation of the 528 compounds found in renal cortex. (**c**) Heat map representation of the hierarchical clustering analysis that was performed of the 25 lipid species with lower p-values obtained by a one-way ANOVA with a post hoc Tukey’s test. Each line represents an accurate mass ordered by retention time, colored by its abundance intensity, normalized to internal standard, and baselined to the median of all the samples. The scale from -4 or -2 (blue) to 4 or 2 (red) represents this normalized abundance in arbitrary units. For all of them, data was acquired in the untargeted lipidomic profiling analysis based on LC-MS with ESI (+).

**Table 1 metabolites-09-00280-t001:** Protein damage markers detected and measured by GC-MS in renal cortex.

Compound	Adult	Aged	MetR
2-SC	69.55 ± 7.41	68.82 ± 7.80	72.58 ± 3.01
GSA	5232.67 ± 458.74	5216.56 ± 178.61	5729.25 ± 376.60
AASA	284.64 ± 37.76	276.84 ± 21.80	291.35 ± 21.36
CEL	384.75 ± 42.28	322.70 ± 38.12	329.99 ± 22.78
MDAL	228.53 ± 25.27	203.44 ± 25.44	204.02 ± 43.33
CMC	103.78 ± 15.42	99.27 ± 9.04	103.43 ± 7.04
CML	1595.83 ± 75.82	1410.95 ± 123.92	1494.69 ± 110.89

Values are mean ± SEM of n = 10 specimens per group. Units: µmol/mmol Lys.

**Table 2 metabolites-09-00280-t002:** Details of the identified lipid and metabolite species in rat renal cortex which displayed statistically significant differences between the Adult and Aged groups.

Compound	Adduct	*m*/*z*	RT	Log FC	p	Aged vs Adult
PC(P-34:2) ^a^	M^+^H^+^	742.5863	7.64	7.83	0.0253	up
MAG(18:0) ^a*^	M^+^Na^+^	381.2909	4.47	13.00	0.00159	up
DAG(32:2) ^a*^	M^+^H^+^-H2O	529.4520	8.41	7.02	0.0389	up
TAG(50:2) ^a^	M^+^NH4^+^	848.7654	9.55	9.02	0.0134	up
TAG(56:8) ^a^	M^+^H^+^	903.7554	9.78	-8.74	0.0372	down
TAG(56:5) ^a^	M^+^NH4^+^	926.8102	10.01	9.15	0.0242	up
Palmitaldehyde ^b^	M^+^NH4^+^	258.2738	9.11	7.20	0.00955	up
EET methyl ester ^b^	M^+^H^+^-H2O	317.2477	11.83	-6.73	0.0473	down
AA methyl ester ^b^	M^+^NH4^+^	336.2908	11.80	-6.85	0.0275	down
Dihomo-PGI2 ^b^	M^+^H^+^-H2O	363.2525	11.47	-9.34	0.00401	down
ADP-ribose ^b^	M^+^H^+^-H2O	542.0597	0.58	-9.47	0.0423	down
NADH ^b^	M^+^Na^+^	688.1027	5.90	-8.42	0.0418	down

^a^ Compounds detected by the lipidomic approach; ^b^ Compounds detected by the metabolomic approach. Results were obtained by a non-parametric t-test of the data acquired using an untargeted lipidomic and metabolomic profiling method based on LC-MS with ESI (+). Identities were confirmed by exact mass, retention time and isotopic distribution. * Compounds identified by MS/MS.

**Table 3 metabolites-09-00280-t003:** Details of the identified renal lipid species with statistically significant differences between the Adult control, Aged control, and MetR groups.

Compound	*m*/*z*	RT	*p* value	LogFC MetR vs Aged	MetR vs Aged	LogFC MetR vs Adult	MetR vs Adult
LysoPE(18:3) *	493.303	0.83	0.0197	10.67	up	5.68	up
LysoPI(15:0)	559.2942	2.97	0.0234	-9.06	down	-0.03	down
PA(P-30:1)	585.4237	6.41	0.00000662	14.26	up	12.35	up
PC(20:0)	604.3236	2.13	0.0128	10.73	up	7.02	up
PC(38:6) *	823.5946	5.58	0.00246	8.39	up	0.07	up
PC(42:2)	870.688	7.40	0.0478	-7.80	down	-7.05	down
PC(P-38:2)	820.6198	8.70	0.00248	-7.57	down	-0.07	down
PG(32:1)	721.5002	7.72	0.0397	0.05	up	-0.12	down
PS(P-29:0)	642.453	5.67	0.03	-5.80	down	1.62	up
GlcAβ-Cer(d36:1)	742.5863	7.64	0.002	-10.15	down	-1.76	down
SM(d30:0) *	666.5661	8.41	0.02	9.08	up	7.20	up
MAG(20:3)	381.2906	4.48	0.00482	-6.16	down	8.82	up
DAG(32:2) *	529.4533	8.41	0.0173	0.61	up	7.37	up
TAG(40:2)	848.7654	9.56	0.00188	-10.11	down	1.85	up
TAG(61:6)	941.8264	9.94	0.0235	11.11	up	1.74	up

The results were obtained by a one-way ANOVA with a post hoc Tukey’s test of the data acquired using an untargeted lipidomic profiling method based on LC-MS with ESI (+). The identities were confirmed by exact mass, retention time, and isotopic distribution. * Compounds identified by MS/MS.

**Table 4 metabolites-09-00280-t004:** Details of the identified renal metabolite species with statistically significant differences between the Adult control, Aged control, and MetR groups.

Compound	*m*/*z*	RT	*p* value	LogFC MetR vs Aged	MetR vs Aged	LogFC MetR vs Adult	MetR vs Adult
Methylcytidine	258.0986	0.48	0.0224	-2.27	down	0.69	up
ADP-ribose	542.0597	0.58	0.0036	-0.02	down	-2.36	down
Inosine	269.0814	0.67	0.0291	0.01	up	-0.12	down
Glutathione disulfide	613.1499	0.46	0.0169	-2.46	down	-0.51	down
Angiotensin (1-7)	937.4213	11.50	0.0375	2.29	up	2.58	up
epoxyHDHA	399.2466	10.04	0.0183	2.62	up	0.48	up
10,11-DiHDPE	363.2511	11.47	0.0236	2.35	up	0.05	up
20-carboxy-LTB4	367.2129	7.33	0.0143	-0.11	down	0.00	down
6-keto PGE1	369.2334	6.89	0.0192	-2.37	down	-0.51	down
5β-Cholestane-3α,7α-diol	427.3541	13.89	0.0242	-0.07	down	1.98	up
Glycocholic Acid	466.3105	8.93	0.000176	2.89	up	2.85	up
Dodecenoyl-CoA	912.2679	13.69	0.0435	-1.18	down	1.03	up
Formyl-CoA	778.0916	10.56	0.0215	0.55	up	-1.75	down
Palmitaldehyde	258.2745	9.11	0.00461	0.01	up	2.69	up
Propionylcarnitine	235.1639	9.19	0.0278	0.00	up	2.36	up

The results were obtained by a one-way ANOVA with a post hoc Tukey’s test of the data acquired using an untargeted metabolomic profiling method based on LC-MS with ESI (+). The identities were confirmed by exact mass, retention time, and isotopic distribution.

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
