# Peer review of "Effects of Aging and Methionine Restriction on Rat Kidney Metabolome"

_metabolites, 2019, doi:10.3390/metabo9110280_

Round 1

Reviewer 1 Report

This manuscript studied age and MetR effects on rat renal cortex metabolomes and lipidomes. Samples were obtained from three groups of rats (adult control group, aged control group and MetR group) and mass spectrometry was used for identification and quantification of lipids and other metabolites. The experiments were carefully designed and the content in this manuscript was generally logically presented, however, there are a few questions needed to be addressed.  

1) Is there a reason data in Figure 1 were baselined to median and data in Figure 2 were baselined to mean?

2) Type in line 107: this to these

3) Clustering of 25 lipid species in both figure 1 and Figure 2 is confusing. Why 25 lipids? Why not use all significantly different features?

These 25 species were from the t-test or ANOVA test. I think describing the results of these tests ahead of the hierarchical clustering of these species is easier to follow.

4) Typo in line 397: antargeted to untargeted

5) Line 181-182: Please add more description for a class-dependent way.

6) It will be good to provide other methods of verification for the metabolites that showed different trend of changes when compared MetR verse aged or adult normal.

Author Response

This manuscript studied age and MetR effects on rat renal cortex metabolomes and lipidomes. Samples were obtained from three groups of rats (adult control group, aged control group and MetR group) and mass spectrometry was used for identification and quantification of lipids and other metabolites. The experiments were carefully designed and the content in this manuscript was generally logically presented, however, there are a few questions needed to be addressed.  

We thank the reviewer for his/her nice comments about our manuscript.

1) Is there a reason data in Figure 1 were baselined to median and data in Figure 2 were baselined to mean?

A: There was an error in Figure legend 2. The correct sentence is the following: “…baselined to median of all samples …”. We have amended this error in the new version of the manuscript.

2) Type in line 107: this to these

A: We thank the reviewer for this appreciation. We have amended this error.

3) Clustering of 25 lipid species in both figure 1 and Figure 2 is confusing. Why 25 lipids? Why not use all significantly different features?

These 25 species were from the t-test or ANOVA test. I think describing the results of these tests ahead of the hierarchical clustering of these species is easier to follow.

A: We apologize for the misunderstanding. As it is now specified in Figure 1C and 2C we selected those 25 molecules with lower p-value (regardless of whether they are statistically significant or not). The selection of 25 instead of 20 or 30 is arbitrary and is has only the purpose of a good visualization. If we include all the statistically significant molecules in one case could be 100 hundred molecules and in the other only 3. We have decided to choose always the 25 with the lowest p value in order to homogenize our results and obtain a good visualization of the data. However, although not all the statistical different molecules were represented in this representation, we included all the data about the metabolites in the supplementary data.

4) Typo in line 397: antargeted to untargeted

A: We thank the reviewer for the appreciation. We have amended this error. The correct word is Untargeted.

5) Line 181-182: Please add more description for a class-dependent way.

A: We apologize for the confusing sentence. We clarify this aspect in the new version of the manuscript:

“Finally, the identified lysoglycerophospoholipids (LGPs) appeared to be regulated in a class-dependent way (being the LPE increased and LPI decreased in MetR group.

6) It will be good to provide other methods of verification for the metabolites that showed different trend of changes when compared MetR verse aged or adult normal.

 A: We agree the reviewer that it will be good to confirm the changes observed using another methodology. Although we confirm the metabolites identified using exact mass, retention time, isotopic distribution and/or MS/MS spectrum alternative techniques should be used in the future to validate our results.  We are working on this aspect for a future paper.

Reviewer 2 Report

In the present manuscript is described the lipodomic and metabolomic profile in Wistar rats comparing adult and older rats with restriction in methionine intake. 

Overall the manuscript is well written. I would suggest to the authors to modify the p value in the tables with the classical p<0.0... and instead on writing up/down, I would prefer using arrows up and down on depending on significance e.g ↓(p<0.05), ↓↓ (p<0.01), ↓↓↓ (p<0.001) or ↑, ↑↑, ↑↑↑ (the contrary).

Moreover it is not clear to me whether the authors have identified specific pathways. In particular, in the discussion part (line 234) the description of the possible metabolic pathways is left too open in my opinion. Which are the metabolic pathways that results more affected, what is the explanation, are there any clinical data that would support the effects seen in animal models to resemble the human physiology? 

Author Response

Comments and Suggestions for Authors

In the present manuscript is described the lipodomic and metabolomic profile in Wistar rats comparing adult and older rats with restriction in methionine intake. 

Overall the manuscript is well written. I would suggest to the authors to modify the p value in the tables with the classical p<0.0... and instead on writing up/down, I would prefer using arrows up and down on depending on significance e.g ↓(p<0.05), ↓↓ (p<0.01), ↓↓↓ (p<0.001) or ↑, ↑↑, ↑↑↑ (the contrary).

A: We thank the reviewer for his/her nice comments and suggestions. Although we agree the reviewer that using arrows the representation of the change is clearer, when we compare more than two groups this representation is not possible. As it can be seen, for example,  in Table 3 and 4 the p value offered is the global p value of ANOVA but the sense of the change could be different in each post-hoc comparison. In order to further clarify the magnitude of the change in each comparison we provide, as well as the global p-value, the Fold Change value.

Example

Compound

m/z

RT

p value

LogFC MetR vs Aged

MetR vs Aged

LogFC MetR vs Adult

MetR vs Adult

PG(32:1)

721.5002

7.72

3.97E-02

0.05

up

-0.12

down

Moreover it is not clear to me whether the authors have identified specific pathways. In particular, in the discussion part (line 234) the description of the possible metabolic pathways is left too open in my opinion. Which are the metabolic pathways that results more affected, what is the explanation, are there any clinical data that would support the effects seen in animal models to resemble the human physiology? 

A: As the reviewer indicates, the sentence (line 234) is a general and open description. We did it this way in order to develop this idea in the following paragraph of the discussion. We have clarified this point in the new version of the manuscript.

Round 2

Reviewer 2 Report

Thanks for answering on my questions and for addressing in the paper my comments.